# Maintenance of Photosynthesis as Leaves Age Improves Whole Plant Water Use Efficiency in an Australian Wheat Cultivar

**Bailey Kretzler [1],\*, Cristina Rodrigues Gabriel Sales [1],† , Michal Karady [2],
Elizabete Carmo-Silva [1] and Ian C. Dodd [1],\***

[1]   Lancaster Environment Centre, Lancaster University, Lancaster LA1 4YQ, UK;
      crisgabi.sales@gmail.com (C.R.G.S.); e.carmosilva@lancaster.ac.uk (E.C.-S.)
[2]   Laboratory of Growth Regulators, Institute of Experimental Botany of the Czech Academy of Sciences and
      Faculty of Science of Palacký University, Šlechtitelů 27, CZ-78371 Olomouc, Czech Republic;
      Michal.karady@upol.cz
\*    Correspondence: kretzlerbailey@gmail.com (B.K.); i.dodd@lancaster.ac.uk (I.C.D.);
      Tel.: +1636-866-6688 (B.K.); +44-(0)1524-593809 (I.C.D.)
†    Present Address: Department of Plant Sciences, University of Cambridge, Cambridge CB2 3EA, UK.

**Abstract:** Leaf-level water use efficiency ($WUE_i$) is often used to predict whole plant water use efficiency ($WUE_{wp}$), however these measures rarely correlate. A better understanding of the underlying physiological relationship between $WUE_i$ and $WUE_{wp}$ would enable efficient phenotyping of this important plant trait to inform future crop breeding efforts. Although $WUE_i$ varies across leaf age and position, less is understood about the regulatory mechanisms. $WUE_i$ and $WUE_{wp}$ were determined in Australian (cv. Krichauff) and UK (cv. Gatsby) wheat cultivars. Leaf gas exchange was measured as leaves aged and evaluated in relation to foliar abscisic acid (ABA) and 1-aminocyclopropane-1-carboxylic acid (ACC) concentration, chlorophyll content and Rubisco activity. Carbon dioxide ($CO_2$) assimilation ($A$) declined more rapidly as leaves aged in the lower $WUE_{wp}$ genotype Gatsby. Both ACC concentration and Rubisco activity declined as leaves aged, but neither explained the variation in $A$. Further, stomatal conductance ($g_s$) and stomatal sensitivity to ABA were unchanged as leaves aged, therefore $WUE_i$ was lowest in Gatsby. Maintenance of $A$ as the leaves aged in the Australian cultivar Krichauff enabled greater biomass production even as water loss continued similarly in both genotypes, resulting in higher $WUE_{wp}$.

**Keywords:** wheat; photosynthesis; leaf age; Rubisco; *Triticum aestivum*; water use efficiency; 1-aminocyclopropane-1-carboxylic acid (ACC); abscisic acid (ABA)

## 1. Introduction

Wheat contributed 20% of the world's calories in 2003 and continues to be one of the most globally significant cereals next to corn and rice [1]. However, wheat's projected production is expected to fall short of estimated consumption levels as the year 2050 approaches [2]. Further, wheat yield and whole plant water use efficiency ($WUE_{wp}$) are negatively impacted by water deficit conditions [3–5], which are becoming more prevalent in many wheat growing regions (e.g., The North China Plain) [6–8]. Intensive agriculture in these regions extracts more groundwater than is replaced through drainage, causing a decline in aquifer water tables, which in turn decreases agricultural sustainability [8]. Thus, it is crucial to identify wheat germplasm with improved $WUE_{wp}$ (biomass gain/water use) with the potential for drought tolerance and water savings. Many studies measure leaf-level water use efficiency ($WUE_i$—carbon dioxide ($CO_2$) assimilation per stomatal conductance) via infrared gas analyzers

(IRGAs). However, $WUE_i$ is infrequently an adequate predictor of $WUE_{wp}$ due to the temporal fluctuations in the leaf level response [9–12]. Therefore, it is imperative to either (1) improve the timing and/or replication of $WUE_i$ measurements, or (2) find alternative measurements (e.g., $\Delta^{13}C$ or hormone profiles) that can predict $WUE_{wp}$ in early vegetative stages. In achieving the latter, biochemical pathways that lead to improved $WUE_{wp}$ should be examined to identify phenotyping tools that streamline breeding for WUE.

Multiple pathways have been proposed to enhance wheat yield and $WUE_{wp}$, including those that increase photosynthetic capacity and/or alter stomatal sensitivity [1,10–16]. Improved photosynthetic capacity can be achieved by increasing the content and/or activity of the photosynthetic enzyme ribulose-1,5-bisphosphate carboxylase/oxygenase (Rubisco), or the rate of ribulose-1,5-bisphosphate (RuBP) regeneration [12–14]. Decline in the initial activity and activation state of Rubisco corresponded with a decline in photosynthetic capacity in wheat grown in high temperatures [17]. Further, the content and activity of Rubisco declined in wheat grown in high temperatures or elevated $CO_2$ levels, and was again correlated to a decline in photosynthetic capacity [18]. Increased Rubisco content and total and initial activity also showed significant correlation to greater maximum and ambient $CO_2$ assimilation rates ($A_{max}$ and $A$, respectively) in field-grown wheat [19]. The maximum regeneration rate of RUBP ($J_{max}$), the substrate carboxylated by Rubisco, can limit photosynthetic capacity in field-grown wheat, where higher $J_{max}$ correlated to a higher $A_{max}$ [19]. Faster rates of RuBP regeneration ($J$) have also been associated with higher Rubisco activation states and $A$ in wheat and rice [20]. The cumulative $CO_2$ assimilation across a crop's lifetime ultimately determines the extent of biomass gain [13]. Therefore, germplasm with lower limitations in terms of Rubisco content and activity and RuBP regeneration will be less likely to have restricted $CO_2$ assimilation rates, and ideally produce more biomass.

Stomatal conductance ($g_s$) can also limit $A$ by regulating the amount of $CO_2$ available for photosynthesis [9,21–23]. Differences in stomatal sensitivity to environmental conditions influence water use as well as $A$ and biomass gain [16,20,23]. Many studies show that stomatal sensitivity to both vapor pressure deficit (VPD) and soil moisture content varies between wheat genotypes [16,20,21,24–26]. Wheat cultivars with higher $g_s$ than traditional varieties had correspondingly higher $A$ and thousand grain weight under well-watered conditions [16]. However, stomatal sensitivity to soil drying varied within both groups, indicating that higher yields may not be directly attributed to stomatal sensitivity [16]. Still, differential stomatal sensitivity among wheat genotypes may be useful in selecting varieties that are best suited for varying drought scenarios [24]. Usually, $g_s$ and hence $A$ vary across the crop canopy due to stomatal patchiness and differences in microclimate [10,27–31]. In some cases, the relationship between $A$ and $g_s$ changes as leaves age, where $A$ declines before $g_s$, thereby decreasing $WUE_i$ [23]. Thus, the cumulative contribution of a leaf to water use and biomass gain will vary across time and position, indicating that momentary measures of $WUE_i$ ($A/g_s$) cannot predict the whole plant response. The complex nature of these relationships suggest that further attention should be paid to how leaf age and environment influence $WUE_i$ and ultimately biomass gain, $WUE_{wp}$, and yield.

This study focused on the impact that leaf age has on $A$, $g_s$, and $WUE_i$, and their relationship to $WUE_{wp}$. $WUE_i$ typically decreases with leaf age as $A$ declines while $g_s$ is maintained, with the latter attributed to reduced stomatal sensitivity to abscisic acid (ABA) [14,23,32]. How this relationship varies between genotypes has not been examined, although variation in stomatal sensitivity to stress-induced ABA has been identified in wheat [33,34]. The decline in $A$ in aging leaves has consistently been associated with higher levels of ethylene or its precursor 1-aminocyclopropane-1-carboxylic acid (ACC), which induces leaf senescence [32,35–37]. Further, chemical inhibition of ethylene synthesis via amino-ethoxyvinylglycine spray increased stomatal conductance and photosynthesis levels in wheat, with the latter being attributed to improvements in photochemical efficiency (i.e., efficiency of PSII and electron transport rate) [35]. ACC/ethylene is also involved in cross talk with ABA, where increases in ACC decreased ABA content or stomatal sensitivity to ABA [38–42]. Additionally, stomatal sensitivity to ABA decreases with leaf age, whereas sensitivity to ethylene/ACC increases with leaf age [23,32]. Thus, as leaves age and senesce, Rubisco activity and $A$ decline before $g_s$, which eventually declines

due to an increase in stomatal sensitivity to ethylene. Both hormones should therefore be further examined for their impact on $g_s$ as well as on $A$, chlorophyll content, and Rubisco activity and content. Presently, few studies exist in wheat where natural changes in the content of and stomatal sensitivity to ABA and ACC across leaf age have been quantified [32].

With one exception [14], the impact of leaf age on $A$ and Rubisco content and activity across multiple wheat genotypes has scarcely been documented. However, such information is important in improving current and developing new phenotyping techniques. Here, the impact of leaf age on $A$, $g_s$, $WUE_i$, Rubisco, ABA and ACC concentration, and chlorophyll content was examined in both controlled and fluctuating environments (i.e., light and relative humidity were minimally regulated). Additionally, changes in $A$ and $g_s$ were examined in relation to biomass gain and water use. Since genotypes varied in their ability to maintain $A$ as leaves age, we hypothesized that those better able to maintain $A$ would have higher $WUE_i$ and $WUE_{wp}$. Further, we suspect that the mechanisms underlying sustained $A$ could be useful early indicators in $WUE_{wp}$ phenotyping.

## 2. Materials and Methods

### 2.1. Experimental Design

Two pot-based experiments were conducted at Lancaster University (Lancaster, United Kingdom) using the cultivars Krichauff (KR) and Gatsby (GA). These genotypes differ in $WUE_{wp}$, where the Australian cultivar KR shows high $WUE_{wp}$ and the UK cultivar GA shows low $WUE_{wp}$ [43]. The first experiment was conducted in a glasshouse (GH), while the second experiment was conducted in a controlled environment (CE) room. Conditions were set to 16 h/24 °C days and 8 h/18 °C nights. Both experiments followed the same procedure, with the exception of the growth environment (Table S1—Supplemental Data). The naturally lit glasshouse received supplemental lighting below 200 μmol m$^{-2}$ s$^{-1}$ with sodium lamps (600 Watt Plantastar by Oram Ltd., Newton-Le-Willows, UK), while the CE room was lit with LED lights (B150 NS1 by Valoya Oy, Helsinki, Finland). Photoperiod light levels in the CE room were consistently maintained above 100 watts/m$^2$, whereas day-time light levels in the glasshouse ranged from 9 watts/m$^2$ to 714 watts/m$^2$. Hourly temperature and VPD were greater, and more variable, in the GH experiment than the CE experiment (Table S1—Supplemental Data). Plants were grown in two-liter (10.5 × 10.5 × 20 cm) pots with a mix of 3:1 commercial compost mix (Petersfield Growing Medium, Leicester, UK) to silver sand. Three seeds were sown into pre-watered pots, and seedlings were thinned to one per pot at one-week post emergence. Eighteen biological replicates of each genotype were cultivated, which were divided into three measurement groups and randomized using the EDGAR program (developed by James K. M. Brown, Cereals Research Department, John Innes Centre, Norwich, UK). Plants received optimal water and were weighed every 1 to 3 days to determine plant water use by change in pot weight.

### 2.2. Gas Exchange Measurements and Leaf Sampling

Measurements began 21 days after sowing (DAS), with the fourth leaf on the main tiller of each plant measured at weekly intervals over a two-week period for $A$, $g_s$, and $WUE_i$ using a LI-6400xt IRGA equipped with a leaf chamber fluorometer (Licor Biosciences, Lincoln, Nebraska, USA). Instrument settings were 400 μmol mol$^{-1}$ CO$_2$, a saturating light level of 1800 μmol m$^{-2}$ s$^{-1}$, a block temperature of 26 °C, a flow rate of 300 μmol s$^{-1}$, the leaf fan set to high, a leaf vapor pressure deficit of 1.8 ± 0.3 kPa and the relative humidity at ambient levels. These conditions were selected to minimize the impact of factors other than leaf age. The leaf chamber inside the fluorometer was 1.6 cm in diameter. Leaf widths were determined during each measurement for area correction of readings in case the leaf did not fill the chamber. Plants were measured between the hours of 8:30 and 13:00 (i.e., 2.5 to 7 h after the start of the photoperiod) in the same pattern each week, based on the EDGAR randomization. The measurements corresponded with recent emergence (Harvest 1, H1-21 DAS), full expansion (Harvest 2, H2-28 DAS), and leaf maturity (Harvest 3, H3-35 DAS) of the fourth leaf and were collected

from the center of the leaf. At each time point, a subset of replicates was sampled for future leaf ABA, ACC and Rubisco assays. Sampling was completed using a cork and razor apparatus that cut 2.5 cm leaf sections (i.e., the same leaf length the LI-6400xt fluorometer head covers), which were snap frozen in liquid nitrogen and stored at $-80$ °C until processing. Samples were consistently collected in the same order, starting 5 cm from the leaf tip to include the center of the leaf. ABA was the first sample collected, closest to the leaf tip, then Rubisco and ACC to follow. Leaf width was measured between samplings in order to calculate the area of the leaf section. In the GH experiment, all plants with leaf 4 intact (not sampled) were measured weekly with a subset being sampled, whereas in the CE experiment, only the plants to be sampled that week were measured.

### 2.3. Biomass Estimation

Following weekly measurements, plants were harvested to determine above-ground dry biomass and soil gravimetric water content (GWC). Above-ground biomass was collected by cutting the plant at the soil line to exclude root biomass, and was dried in an oven at 60 °C until constant weight. The soil (including the roots) was weighed, then dried in an oven at 105 °C until constant weight. Soil moisture was determined gravimetrically as the experimental pot weight minus the empty pot and dry soil weights, divided by the dry soil weight. In the CE experiment, 4 additional replicates of each genotype were grown alongside the experimental replicates. These were used to calculate average above-ground dry biomass at the fourth leaf stage. Biomass gain in the CE experiment was thus determined by subtracting this average from the final above-ground dry biomass. Following the leaf age measurements in the GH, half of the plants were harvested to determine above-ground dry biomass, while the others continued to grow for 3 additional weeks, after which plants were similarly harvested and dried. Biomass gain in the GH was determined across these 3 weeks.

### 2.4. ABA, ACC, and Rubisco Assays

Leaf samples for ABA quantification were freeze-dried and shaken with deionized water (1:50 extraction ratio) overnight at 4 °C to obtain the leaf extract. ABA concentrations of the extracts were quantified using a radio-immuno assay [44] with 50 μL of either extract or ABA standard (0, 62.5, 125, 250, 500, 1000, 2000 pg ABA per 50 μL solution) being combined with PBS, [$^3$H]-ABA, and MAC 252 antibodies. Samples were incubated for 45 min at 4°C, then saturated ammonium sulfate was added for 30 min. Precipitated samples were centrifuged, the supernatant decanted and the pellet re-suspended in 50% ammonium sulfate, with the procedure repeated to remove any unbound radioactivity. The remaining pellet was then re-suspended in 100 μL of deionized water and mixed with Ecoscint H scintillation fluid to determine radioactivity (CPM) with a scintillation counter. Extract ABA concentrations were determined from a standard curve generated using a logit transformation.

Frozen fresh leaves for ACC analysis were sent to Palacky University (Olomouc, Czechia) on dry ice, where they were extracted via liquid–liquid extraction and processed via mass spectrometry. Briefly, 1 mL of $H_2O$:methanol:chloroform (1:2:1) as extraction solution, 5 pmol of labeled d$^4$-ACC and two zircone beads were added to each frozen sample, which were then homogenized by bead mill for 6 min at 27 Hz (MixerMill, Retsch GmbH, Haan, Germany). Samples were centrifuged (15 min, 15,000 RPM), then 100 μL of supernatant was placed in a 100 μL glass insert in a glass vial for subsequent derivatization, mass spectrometry analysis and quantification. The LC-MS/MS system was comprised of a 1260 Infinity II LC System coupled to a 6490 Triple Quad LC/MS System with Jet Stream and Dual Ion Funnel technologies (Agilent Technologies, Santa Clara, CA, USA). Quantification was performed in Mass Hunter vB.09 (Agilent Technologies) [45]. Values were then quantified by reference to a standard curve and analyzed using RStudio [46].

Rubisco initial and total activity were determined in order to estimate operational activity (initial activity, before in vitro carbamylation) and capacity (total activity, after in vitro carbamylation) [17]. Measuring both initial and total activities provides insight into whether declines in activity are attributable to reduced activation state (i.e., reduced initial but not total) or content (i.e., reduction

in both) [19]. Rubisco samples were processed via a previously described spectrophotometric assay using the coupled enzymes pyruvate-kinase and lactate-dehydrogenase, to determine initial and total activity of the enzyme [47,48]. Part of the extract used for the Rubisco assays was also used to extract chlorophyll using ethanol as an organic solvent, as described in Carmo-Silva et.al. (2008), and was quantified via the methods described by Wintermans and de Mots (1965) with revised equations from Lichtenthaler and Buschmann (2001) [49–51].

## 2.5. Data Analysis

Correlations between measurements were tested using Pearson's correlation. Relationships between correlated parameters were then examined through linear regression modeling. Factors such as genotype and leaf age were included as interactions in the models. These models were tested for significance using a series of analysis of variance (ANOVA) and analysis of covariance (ANCOVA) tests via the R package "car" [52]. Variation between genotypes and leaf ages was also determined using these variance tests. The magnitude of variation was determined using a Tukey post-hoc test with the R package "emmeans" [53]. The emmeans package determined the mean and standard error of each leaf age and genotype combination, allowing these to be ranked accordingly. A significance bracket of 95% was used to determine statistical significance. Graphs to illustrate these relationships were generated using the R package "ggplot2" [54].

## 3. Results

### 3.1. Whole Plant Response

Both wheat cultivars Krichauff (KR) and Gatsby (GA) used more water in the longer GH experiment (over 48 days, reproductive at harvest) than in the CE experiment (over 25 days, vegetative at harvest), but there were no genotypic differences in water use (Figure 1a). However, KR gained 82% more biomass than GA in the GH experiment, with no significant differences noted in the CE experiment. While the $WUE_{wp}$ of KR was approximately twice that of GA in the GH experiment, no significant differences in $WUE_{wp}$ were detected in the CE experiment (Figure 1b). The two genotypes also showed no significant difference in the slopes of the linear relationship between water use and biomass gain (Figure 1a). Such evidence indicates that variation in $WUE_{wp}$ was not detected at the early stages of development, but KR becomes more water-use efficient at the end of the vegetative stage and start of the reproductive stage.

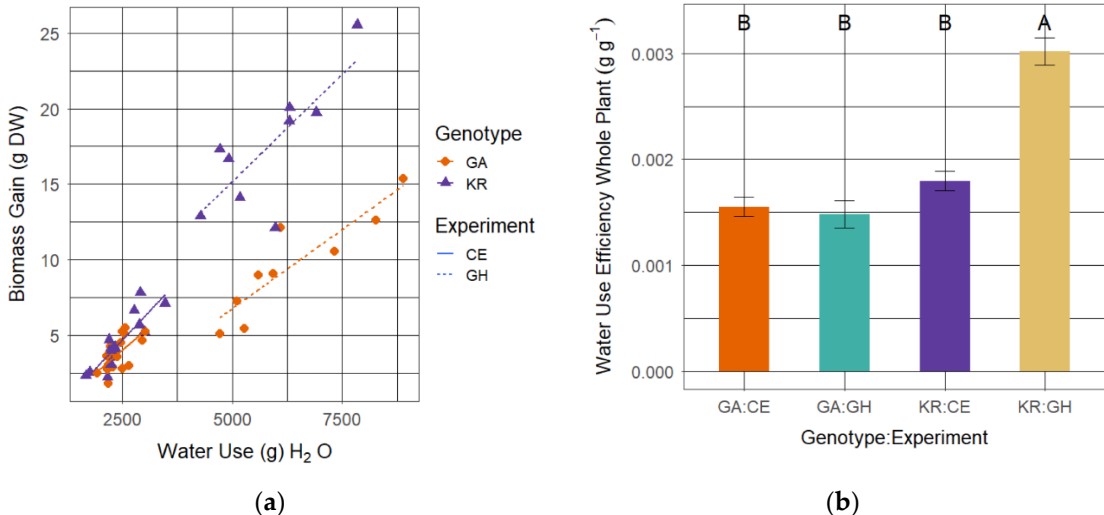

**Figure 1.** The relationship between aboveground biomass gain and water use determined by: (**a**) the linear relationship (BM ~ WU) between biomass gain (BM) and water consumption (WU), where each point is a single biological replicate ($R^2$ = 0.94, $p$ < 0.001) and (**b**) whole plant water use efficiency ($WUE_{wp}$) across two experiments (CE, Controlled environment; GH, Glasshouse) and two wheat cultivars (KR, Krichauff; GA, Gatsby), where values represent means ± SE of 6 biological replicates.

### 3.2. Leaf Level Response across Leaf Age

The wheat cultivars KR and GA showed similar $WUE_i$ responses, as the relationship between $A$ and $g_s$ (Figure 2) was not significantly affected by genotype, leaf age, or the plant growth conditions. The two genotypes assimilated $CO_2$ at comparable rates (no significant difference in $A$), however rate of decline in $A$ as leaves aged differed between genotypes and experiments (Figure 3a). $CO_2$ assimilation ($A$) of GA declined by 31% in week 3 (H3) in the GH experiment and by 30% in week 2 (H2) in the CE experiment, whereas KR showed no significant decline as the leaves aged. Stomatal conductance ($g_s$) did not vary between leaf ages but that of KR was significantly higher than that of GA regardless of leaf age or experiment (Figure 3b). The marginally significant ($p$ = 0.059) decline in $g_s$ of GA from week 1 to 2 (H1 to H2) in the CE experiment would explain the corresponding decline in $A$ (Figure 3a,b). Conversely, in all other cases, $A$ tended to decline with leaf age while $g_s$ did not, thus reductions in $A$ as leaves age are less attributable to diffusional limitations. Regardless, $A$ declines before $g_s$ as leaves age and this decline is delayed in KR.

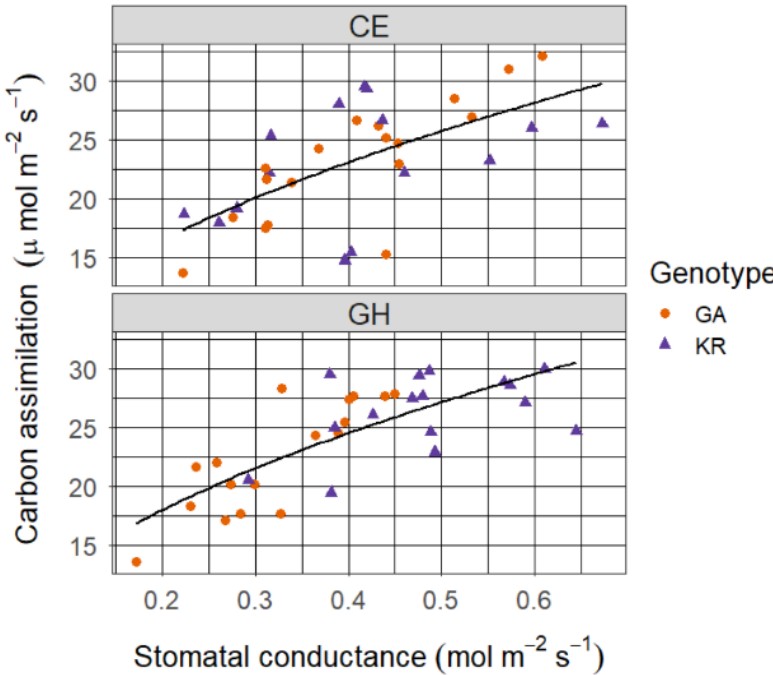

**Figure 2.** The same curvilinear relationship ($A \sim \sqrt{(g^s)}$) between leaf-level $CO_2$ assimilation ($A$) and stomatal conductance ($g_s$) was consistent across two wheat cultivars (KR, Krichauff; GA, Gatsby) grown in the glasshouse (GH) or controlled environment (CE) chamber. Gas exchange was measured at 400 µmol mol$^{-1}$ $CO_2$, PPFD = 1800 µmol m$^{-2}$ s$^{-1}$, and 26 °C (block temp). Each data point represents a single biological replicate and points are colored by genotype. $R^2 = 0.78$, $p < 0.001$.

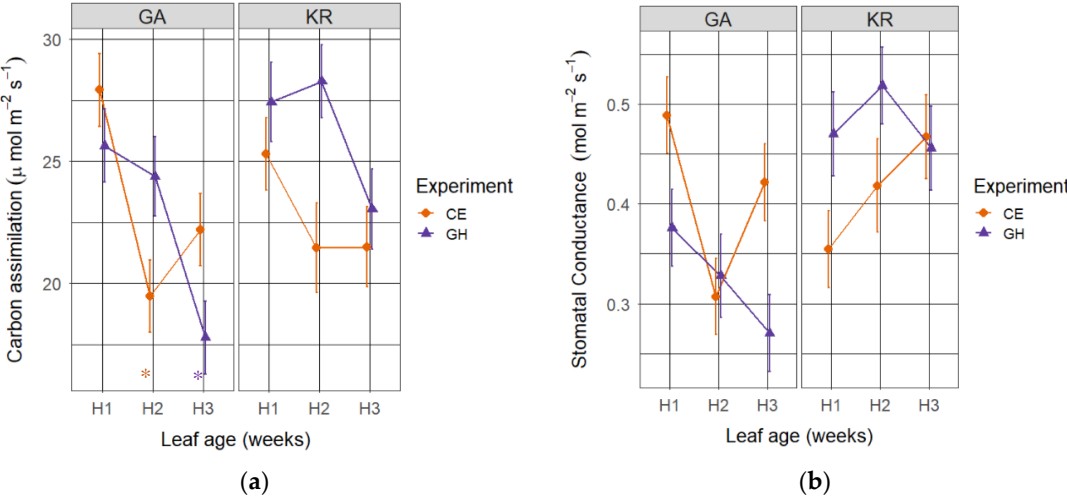

(**a**)                                                                 (**b**)

**Figure 3.** The impact of leaf age on (**a**) $CO_2$ assimilation ($A$) and (**b**) stomatal conductance ($g_s$) in two wheat cultivars (KR, Krichauff; GA, Gatsby) grown in the glasshouse (GH) or controlled environment (CE) chamber. Orange and purple asterisks indicate significant difference from harvest 1 (H1) in the CE and GH experiments, respectively. Gas exchange was measured at 400 µmol mol$^{-1}$ $CO_2$, PPFD = 1800 µmol m$^{-2}$ s$^{-1}$, and 26 °C (block temp), in young expanding (H1-21 DAS), fully expanded (H2-28 DAS) and mature (H3-35 DAS) leaves. Data points represent the means ± standard errors of 6 biological replicates.

### 3.3. Leaf ABA and ACC Concentrations

Foliar ABA concentrations did not differ across genotype or leaf age (Table 1), with no significant relationship between ABA and soil gravimetric water content (GWC) or $g_s$, as expected in well-watered plants (Figure S3–S7, Supplementary Materials). Such findings indicate no genotypic differences in

stomatal sensitivity to foliar ABA as the leaves aged. Foliar ACC concentrations did not vary between genotypes, but changed with leaf age. Notably, foliar ACC concentration of KR significantly declined in the CE experiment between leaf age H1 to H2 and H3 (Table 1, Figure 4). Further, in the GH experiment, foliar ACC concentration tended to decline ($p < 0.07$) from leaf age H1 to H3 in both genotypes. The decline in ACC content was more pronounced in KR than in GA (Figure 4). However, foliar ACC concentration showed no significant relationship with A or $g_s$. Moreover, the ratio of ACC to ABA also showed no difference between genotype or leaf age, and these parameters showed no significant correlation to A or $g_s$. Thus, changes in leaf gas exchange seemed potentially independent of leaf hormone status.

**Table 1.** Mean values of 1-aminocyclopropane-1-carboxylic acid (ACC) and abscisic acid (ABA) concentrations, and their ratio in leaves of two wheat cultivars (KR, Krichauff; GA, Gatsby) grown in the glasshouse (GH) or controlled environment (CE) chamber. Asterisks indicate significant difference from H1 in the relevant experiment and genotype. Samples were collected after completing gas exchange measurements of young expanding (H1-21 DAS), fully expanded (H2-28 DAS) and mature (H3-35 DAS) leaves. Values represent the means ± standard errors of 3–6 biological replicates.

| Genotype | Experiment | Leaf Age | ACC (pg mg$^{-1}$ FW) | ABA (ng g$^{-1}$ DW) | Ratio (ACC/ACC) |
|---|---|---|---|---|---|
| Krichauff | CE | H1 | 147 ± 10 | 238± 102 | 4.1 ± 5.1 |
| | | H2 | 17 ± 12 * | 307 ± 125 | 0.03 ± 6.3 |
| | | H3 | 18 ± 18 * | 34 ± 112 | 11± 8.8 |
| | GH | H1 | 91 ± 13 | 328 ± 112 | 1.6 ± 6.3 |
| | | H2 | 86± 10 | 265 ± 102 | 16 ± 5.1 |
| | | H3 | 28 ± 13 | 227 ±112 | 0.2 ± 6.3 |
| Gatsby | CE | H1 | 33 ± 10 | 258 ± 102 | 2.8 ± 5.1 |
| | | H2 | 10± 10 | 210 ±112 | 0.2 ± 5.1 |
| | | H3 | 11.0 ± 10 | 328 ±102 | 0.02 ± 5.1 |
| | GH | H1 | 88 ± 10 | 168 ± 102 | 1.4 ± 5.1 |
| | | H2 | 67 ± 10 | 325 ± 112 | 1.01 ± 5.1 |
| | | H3 | 37 ± 10 | 296 ± 102 | 0.3 ± 5.1 |

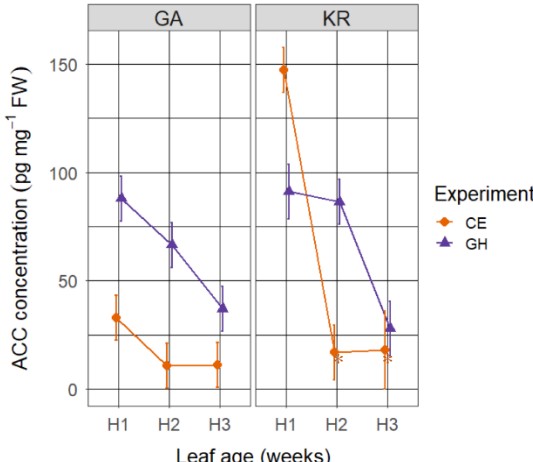

**Figure 4.** The impact of leaf age on 1-aminocyclopropane-1-carboxcylic acid (ACC) concentration in leaves of two wheat cultivars (KR, Krichauff; GA, Gatsby) grown in the glasshouse (GH) or controlled environment (CE) chamber. Orange asterisks indicate significant difference from H1 in the CE experiment. Samples were collected after gas exchange measure were complete from young expanding (H1-21 DAS), fully expanded (H2-28 DAS) and mature (H3-35 DAS) leaves. Data points represent the means ± standard errors of 3 biological replicates.

### 3.4. Leaf Age, Rubisco Activities, and Chlorophyll Content

The two cultivars had comparable rates of initial and total Rubisco activities (Figure 5). Rubisco total activity declined markedly for GA between weeks 2 and 3 (H2 to H3) in the CE experiment, with a similar (but not significant) decline with leaf age also seen in KR. Rubisco initial activity showed a marginal (insignificant) decline in both genotypes indicating no impact on activation state. This decline in Rubisco was not apparently correlated to *A* (Supplementary Figures S1 and S2). Still, decreased Rubisco total activity coincides with decreased $CO_2$ assimilation (cf. Figures 3a and 5), with *A* sustained when Rubisco activities decline less as leaves age (e.g., KR).

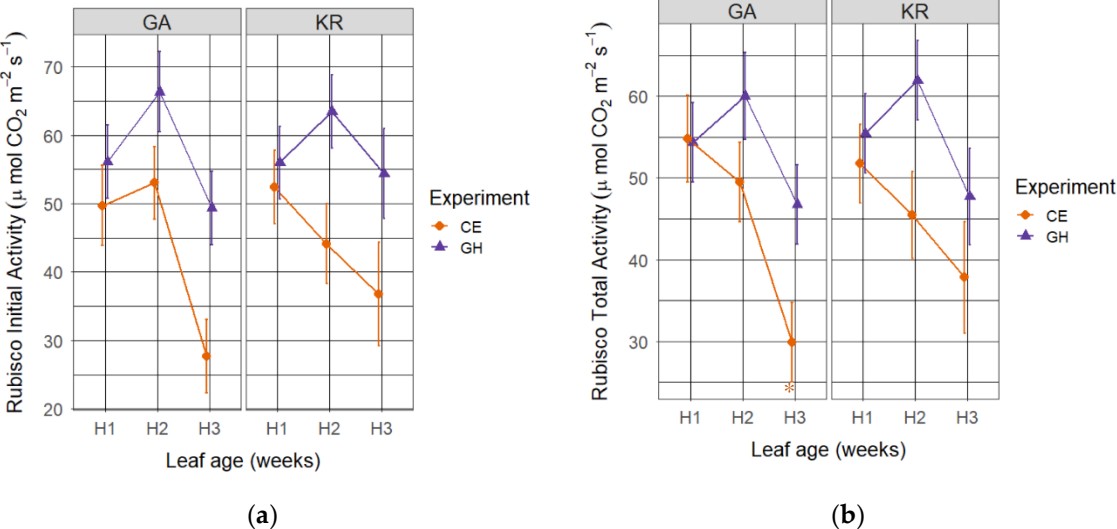

(**a**)  (**b**)

**Figure 5.** The impact of leaf age on Rubisco initial (**a**) and total (**b**) activities in two wheat cultivars (KR, Krichauff; GA, Gatsby) grown in the glasshouse (GH) or controlled environment (CE) room. The orange asterisk indicates significant difference from H1 in the CE experiment. Samples were taken from young expanding (H1-21 DAS), fully expanded (H2-28 DAS) and mature (H3-35 DAS) leaves after gas exchange measurements were completed under growth conditions. Data points represent the means ± standard errors of 6 biological replicates.

Chlorophyll A + B content did not vary with genotype, but actually increased as leaves of KR and GA aged in both experiments. Additionally, $CO_2$ assimilation significantly declined as chlorophyll A + B content increased (Figure 6, but this relationship was unaffected by genotype, leaf age, and experiment.

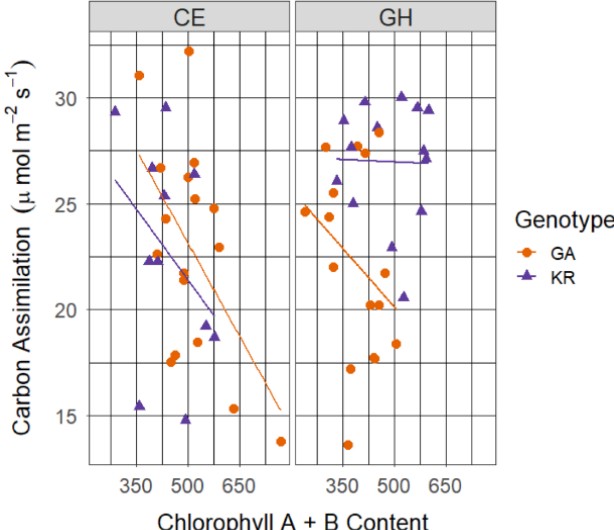

**Figure 6.** The impact of chlorophyll A + B content on $CO_2$ assimilation rate (*A*) for two wheat cultivars

(KR, Krichauff; GA, Gatsby) grown in the glasshouse (GH) or controlled environment (CE) room. Samples were taken from young expanding (H1-21 DAS), fully expanded (H2-28 DAS) and mature (H3-35 DAS) leaves after gas exchange measurements were completed. Data points represent a single biological replicate, of which there were 6 for each genotype and harvest. Lines represent the line of best fit derived from the linear relationship between chlorophyll content and $CO_2$ assimilation (*A* ~ Chlorophyll A + B). $R^2$ = 0.75, $p < 0.001$.

## 4. Discussion

Despite no genetic variation in $WUE_i$ (Figure 3), genotypic differences in $WUE_{wp}$ (2× higher in KR–Figure 1b) were consistent with previous studies in wheat and associated with maintenance of *A* as leaves aged [3–5,43,55]. Further, the 8% increase in $WUE_{wp}$ of KR between the GH experiment (harvested while reproductive), and the CE experiment (harvested while vegetative) indicates ontogenetic variation as evident in wheat and rice [4,56]. KR had higher $WUE_{wp}$ because it gained more biomass than GA despite similar water consumption. At the leaf level, despite similar *A* in young leaves, KR sustained *A* longer than GA as leaves aged, indicating how leaf processes influence whole plant phenotype. An earlier decline in *A* without a corresponding decline in $g_s$, as previously reported in wheat, causes profligate water use without additional biomass gain (e.g., GA) [23]. While genetic variation in the rate of decline in *A* as flag leaves age occurs in wheat, to our knowledge such an effect has not been documented in early vegetative wheat leaves [19]. Since flag leaf *A* correlates with yield, such measurements in vegetative stages may enable early yield predictions [19,57]. Further, early vegetative biomass is utilized as a carbon source for grain filling in reproductive stages, thus leaf level measures during this period may more closely approximate other yield parameters (i.e., nutrient content, grain fill and weight, etc.). Therefore, earlier selection based on vegetative leaf gas exchange could increase pre-breeding throughput by eliminating genotypes with low *A*. In doing so, it seems important to understand the regulation of *A* and $g_s$ as leaves age

Maintenance of $g_s$ while *A* declines in ageing leaves has previously been attributed to partial stomatal insensitivity to ABA, with young fully expanded leaves retaining normal stomatal closure in response to exogenous ABA application [23,32]. Observations here indicated that leaf ABA concentrations of well-watered plants did not vary consistently enough to influence stomatal responsiveness (Table 1). Likewise, $g_s$ and *A* also showed no relationship with ACC or the ratio of ACC to ABA, contrary to previous works, likely because the well-watered conditions constrained variation in phytohormone concentrations [32,35–37]. In contrast, leaf ABA/ethylene ratio explained genetic variation in shoot growth of different wheat cultivars exposed to mild soil drying, suggesting

the interaction of these hormones may be more important in regulating vegetative growth [58]. Independently of any interaction with ABA, a greater reduction in foliar ACC concentration in KR than GA (19% and 33% in the GH and CE experiments, respectively) corresponded with maintenance of *A* in KR as leaves aged (Figures 3a and 4), indicating that ACC may still impact *A* and thus biomass gain.

Decreased chlorophyll content during water deficit-induced leaf senescence has partially explained the decline in *A* [16,43,59]. In contrast, *A* declined as chlorophyll content increased by 11–38% from leaf age H1 to H3 (Figures 3a and 6). Shading can enhance chlorophyll content, which then declines at a slower rate than *A* during natural (age-related) senescence [60,61]. Further, chlorophyll content increased with leaf age and shading, where the increase with leaf age was observed in a summer greenhouse similar to the GH experiment here [62–65]. Artificial leaf shading increased flag leaf chlorophyll content corresponding with a decline in *A*, where increased chlorophyll was interpreted as a compensatory mechanism for light capture and dissipation [63]. Further, it has been proposed that soybean plants over-invest nitrogen in chlorophyll pigmentation, as reduced chlorophyll mutants show enhanced leaf level *A* [65]. Thus, inevitable shading from the upper canopy as the fourth leaf aged may have resulted in increased chlorophyll content (an over investment of nitrogen) at the expense of Rubisco activity and content. Furthermore, significantly reduced Rubisco total activity and marginally reduced Rubisco initial activity as leaves aged likely indicates a decline in Rubisco content and not activation state, as previously reported in wheat flag leaves [19]. These declines coincided with a decline in *A* in the genotype GA (Figures 3a and 5b). Therefore, it is possible that shade-induced changes in nitrogen allocation (increased chlorophyll content but decreased Rubisco content) indirectly limited *A*.

Variation in environmental conditions such as temperature, VPD and light levels between measurement days would further alter the relationship between *A*, $g_s$, ABA, and Rubisco. For example, increased VPD should cause stomatal closure and thus potentially lower *A* regardless of Rubisco activity [25,26]. A greater range of temperature, VPD, and light levels could have resulted in GA's apparently lower $g_s$ in the GH experiment due to greater sensitivity to environmental factors, as previously noted [16,20,21,24–26]. Further, differences in these measurements across the canopy are inevitable and would have changed according to the leaf's local environment as plants gained biomass during the experiment [15]. Thus, future experiments should measure multiple leaves within the canopy to determine the impact of both leaf age and position. Environmental conditions such as canopy light interception, VPD, and leaf temperature should be closely monitored to account for any unexpected variation in canopy microclimate and how this may impact diurnal responses in *A* and $g_s$. It would also be crucial to determine how much longer KR is able to sustain *A* and if this corresponds to changes at the biochemical level (e.g., chlorophyll content, Rubisco activities and content and ABA and ACC concentration). Lastly, it is important to note that we measured two genotypes grown in specific environments, and these methods should be repeated across a greater range of genotypes and conditions (i.e., field conditions) to better understand the impact that leaf level variation has on the whole plant phenotype.

## 5. Conclusions

As leaves aged in the vegetative (pre-booting) stage, leaf $CO_2$ assimilation declined at different rates in two wheat cultivars. Ability to maintain *A* was attributed to sustained Rubisco activity and greater decline in foliar ACC concentration, as seen in the genotype Krichauff over a two-week period. Further, both cultivars increased their chlorophyll content as leaves aged, possibly as a strategy to enhance light capture as these leaves become shaded [64]. Maintenance of $g_s$ as leaves aged indicates excess water use without complementary carbon gain in the cultivar Gatsby. The genotype that sustained *A* for a longer duration (KR) tended to produce more biomass per water consumed and had higher *WUE*$_{wp}$. Variation in the leaf level response of genotypes indicates that momentary measures of *WUE*$_i$ cannot account for the cumulative contribution of leaves to the whole plant phenotype across

time and space. While further investigation is needed, sustainability of *A* can serve as an alternative (if labor intensive) method for predicting $WUE_{wp}$.

**Supplementary Materials:** The following are available online at http://www.mdpi.com/2073-4395/10/8/1102/s1, Table S1: Compiled Data, Figure S1: $CO_2$ assimilation × Rubisco Initial Activity, Figure S2: $CO_2$ assimilation × Rubisco Total Activity, Figure S3: Gravimetric water content × leaf age, Figure S4: ABA concentration × gravimetric water content, Figure S5: Stomatal conductance × Gravimetric water content, Figure S6: Stomatal Conductance × ABA content, Figure S7: Abscisic acid × leaf age.

**Author Contributions:** B.K., C.R.G.S., E.C.-S., I.C.D. designed the experiments, B.K. performed research and data analysis, M.K. performed ACC analysis, B.K. wrote the manuscript with contributions from all authors. All authors have read and agreed to the published version of the manuscript.

**Funding:** This research was funded by Friends of Lancaster University in America, as a part of the funding received for the first author's MSc undertaken at Lancaster University. E.C.-S. acknowledges funding from the UK Biotechnology and Biological Sciences Research Council (BBSRC) under grant number BB/L011786/1. MK acknowledges that the work was supported by The Czech Foundation Agency via GJ20-25948Y and an ERDF project entitled "Development of Pre-Applied Research in Nanotechnology and Biotechnology" (No. CZ.02.1.01/0.0/0.0/17_048/0007323).

**Acknowledgments:** The authors thank Samuel Taylor for training and useful discussions relating to physiological analysis. Additionally, the authors would like to thank Jaime Puertolas for training relating to hormone assays (specifically ABA). We are further grateful for the contributions of Denise Patricia Deza Montoya and Diego Zamudio Ayala (Universidad Nacional Agraria La Molina, Peru), who assisted with data collection as part of a Newton Funded Institutional Links grant (to I.C.D.).

**Conflicts of Interest:** The authors declare no conflict of interest.

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
