# Peer review of "Maintenance of Photosynthesis as Leaves Age Improves Whole Plant Water Use Efficiency in an Australian Wheat Cultivar"

_agronomy, doi:10.3390/agronomy10081102_

Round 1
Reviewer 1 Report
The authors set out to investigate the relationship between leaf level and whole plant water use efficiency in two wheat cultivars at the physiological level with the aim to identify phenotyping parameters. They investigated the relationships between leaf age/developmental stage, genotype, stomatal conductance/ sensitivity (ABA, ACC), C assimilation rates, chlorophyll content and biomass gain. Their main finding established that the cultivar with higher whole plant WUE was able to produce more biomass by maintaining higher photosynthetic rates at later developmental stages without using more water. The authors concluded that analyses of more physiological parameters/conditions are needed to identify a rapid and simple screening parameter for high whole plant WUE, but C assimilation can serve as a proxy albeit more labour intensive.
The manuscript is well written, data are presented clearly and experimental design is well thought-through. I have a few points that may warrant clarification or further discussion.
Results
l.76 on: It is not quite clear how the whole biomass was measured. Form the methods it sounds as if the whole plants were harvested for biomass but the number of plants planted seems not enough to warrant that many data points. If only part of the plant was harvested for biomass then this needs to be clarified more. Also there is the caveat that biomass gain is not necessarily even out across the plant leaves, and changes again when taking grain filling into account. It also should be pointed out that underground biomass was not included which may – particularly when it comes to water use be of relevance in terms of mass, fine root development, and root architecture and rooting depth.
As to Figures, just to note that the colours do not come out well on grayscale. Axis ticks and gridlines are barely visible.
l.193 on (Fig. 2): The overall message from Fig. 2 seems to be that A/gs is close to linear in CE grown plants. In fact, it may be worthwhile to separate the two genotypes. GA seems to have a more strict linear correlation. The situation in GH grown plants seems to differ. GA a and KR forms almost two separate pools (linearity within each pool is debatable) with GA operating at low A low gs and KR at high A high gs. Clearly CE and GH differ. What is the reason? Also, Fig. 2 seems to be inconsistent with a decline of A before gs, which would result in a much stronger condensation of different A values over the same gs. Is there an explanation?
I feel the differences between CE and GH plants could be discussed further in regarding to different responses of the two genotypes (also apparent from Fig. 3).
l.245 on: What is the rationale behind measuring the initial Rubisco activity? It is not discussed further and should be a supplement.
l.258: In the context of Chla+b content, it would be god to a) give more information about light levels within the canopy, and in the Glass house diurnally. Lower A with higher Chla+b certainly could be a consequence of shading. But it also raises the question whether Rubisco became RUBP regeneration-limited. Is that what ‘detrimental’ refers to (discussion l.313)? This needs further explanation in the discussion.
The study as a whole is a bit vague about the effects of growth light, which is a major player in determining A. Particularly during midday it is possible that A can be severely diminished due to photoprotection/photoinhibition. Carotenoid profiles and Chl fluorescence measurements would have been good complements to gas exchange A time course of A on certain days would have been helpful, together with tracking of incident light (within and outside the canopy), which is to some extent acknowledged in the discussion.
As end result total biomass is the integration of all A values (-respiratory C losses) across the life time of the plant, modulated many environmental factors. This study was able to identify A as a driving factor but needs further examination of which environmental and developmental factors drive A.
In the conclusion, sustainability of A was suggested as measure for WUEwp. It may be worthwhile to point out, this as a single case study and different plants, varieties, genotypes, environmental conditions, field vs controlled environment may or may not support this.
Reviewer 2 Report
The work is well introduced. My only concern is the lack of hypothesis.
Materials and Methods section in generally well presented, however ABA quantification and Rubisco assays may have been described to shortly. Some modifications probably have been made when compared to original (cited) assays.
Results section: ACC is not a phytohormone, so 3.3 sections should be renamed. All results that statistically differ should be clearly presented in all pictures/tables (asterisks or in other way).
I have no concerns regarding Discussion.
Conclusions should be more general.
